# Study on Increasing the Binding Amount of Rubber and Reinforcing Filler by Adding Aromatic Solvent Oil

**DOI:** 10.3390/polym14132745

**Published:** 2022-07-05

**Authors:** Xiaoqing Li, Zhonghang Fang, Xinmin Shen, Qin Yin, Zhiyuan Chen, Qunzhang Tu, Ming Pan

**Affiliations:** College of Field Engineering, Army Engineering University of PLA, Nanjing 210007, China; lxiaoqing0507@163.com (X.L.); dafengyinqin@126.com (Q.Y.); chenzhiyuan2333@126.com (Z.C.); tqzlhnj@126.com (Q.T.); panlvnn@163.com (M.P.)

**Keywords:** rubber, aromatic solvent oil, reinforcing filler, binding amount, curing characteristics, physical performance, tensile properties

## Abstract

The binding amount of rubber and reinforcing filler directly affects the quality of rubber products. The effect of aromatic solvent oil (S-150) on the binding amount of rubber and reinforcing filler was studied. In order to determine the suitability of rubber after adding S-150, the curing characteristics, physical performance and tensile properties of rubber samples were tested and analyzed. Meanwhile, the microstructure of the composite was analyzed by scanning electron microscopy (SEM). The test results showed that the binding amount of rubber and reinforcing filler was increased after adding S-150. The density and Shore A hardness were decreased. When carbon black was 80 phr, after adding 40 phr of S-150, the rebound resilience of rubber increased by 13% on average, and the elongation at break increased by 88% on average. When white carbon black was between 10-70 phr, after adding 65 phr of S-150, the rebound resilience of rubber increased by 9% on average, and the elongation at break increased by 51% on average. Modulus at 100% and tensile strength were decreased. Meanwhile, it could be judged from the microstructure results that the reticulation space inside the rubber was increased, the agglomerate particles were relatively uniform, and no bubbles or holes were observed. The mechanism that S-150 could increase the binding amount of rubber was analyzed according to the like-dissolves-like principle. This research achievement could lead to improvements in the quality of rubber products and promote their practical application.

## 1. Introduction

Rubber products are used in a variety of applications, such as sealing rings [1], rubber hoses [2], shoes [3], tires [4], etc. Reinforcing filler can constitute a large proportion of rubber formulation [5]. The amount of filler used is important. It can not only improve the physical and mechanical properties of rubber products, but also improve the processability of rubber to a certain extent. The cost of rubber products is directly related to filler. Therefore, testing the effect of different binding amounts of rubber can be consequential [6,7,8]. Unfortunately, the binding amount of rubber and reinforcing filler is limited, which limits its further application [9]. Therefore, a key factor for improving the comprehensive performance of composite materials is the improvement of the binding amount of rubber and reinforcing filler.

There are many types of reinforcing filler in rubber, mainly including carbon black (CB) [10], white carbon black (WCB, also be called as Silica) [11], metal oxides [12], some organic compounds [13], rubber powder [14], recycled rubber [15], and short fibers [16]. Shi et al. [17] investigated the Payne effect of CB-filled natural rubber (NR) compounds under large strain amplitudes, and proved that low temperature and high frequency accelerate the Payne effect. Nafise et al. [18] had displayed improvements in the tribological behavior of both modified WCB-filled curing over pristine CB- and WCB-filled curing. He et al. [19] showed that the mechanical properties of silicone rubber (SR) were obviously enhanced with an increase in WCB amount. CB and WCB are indispensable reinforcing filler, which cannot be replaced at present [20,21]. At the same time, there are many kinds of rubber, such as NR [22], styrene butadiene rubber (SBR) [23], butadiene rubber (BR) [24] and SR [25]. CB and WCB are used as reinforcing filler to reinforce NR and SBR [26,27,28]. Choi et al. [26] investigated the variation of properties of CB- or WCB-filled compounds with varying filler amounts. Shore A hardness, modulus, tensile strength, and wear property were improved gradually by increasing the filler amount. Rattanasom et al. [27] studied the reinforcement of NR with CB/WCB hybrid filler at various ratios in order to determine the optimum CB/WCB ratio, and the results revealed that a curing containing 20 and 30 phr of CB in hybrid filler exhibited the better overall mechanical properties. Sirisinha et al. [28] found the properties of tire tread compounds based on functional SBR and functional NR. The results suggest a significant enhancement in the properties of tire tread compounds from the presence of active functional groups in NR and solution styrene–butadiene rubber (SSBR) molecules.

At present, there are many methods to improve the performance of rubber in industrial production. Changing the rubber preparation process is one method to ameliorate rubber properties [29,30,31,32,33,34]. Zhu et al. [31] developed continuous wet mixing technology. It was found that the continuous wet mixing technology significantly improved the properties of rubber compounds compared with the conventional dry mixing method. The proposed method not only has reasonable processing properties, but also significantly improved the physical properties, compressive fatigue, degree of dispersion and dynamic mechanical properties of the compounds compared with those obtained from the conventional dry mixing process. The search for new types of reinforcing filler is also one of the hot topics in current research [32,33,34,35]. Masłowski et al. [35] developed a novel filler. This filler contained barley, corn and wheat straw. The rubber mixtures containing lignocellulosic materials demonstrated favorable characteristic kinetics for cross-linking. The addition of filler in an appropriate amount to modify NR could improve mechanical and barrier properties of composites. Rubber and filler were modified to improve the properties of rubber [36,37,38]. Miedzianowska et al. [39] studied the functional properties of elastomer composites reinforced with modified lignocellulosic material obtained from cereal straw. Adding oil to rubber can also improve the properties of rubber [40,41,42]. Zhou et al. [42] developed a fluorescence microscopy that showed bio-oil was successfully absorbed by rubber particles, significantly improving the dispersion of rubber in asphalt.

Although there are many ways to improve the properties of rubber, there are few studies on how to improve the binding amount of rubber and filler. Based on the above problems, aromatic solvent oil (S-150) was added to the rubber formula in this study, hoping to improve the binding amount of rubber and filler. In order to verify the suitability of S-150 in rubber formulation, CB and WCB were selected as reinforcing filler, and NR and SBR were used as the rubber matrix. The effect of constant ratio and that of a constant amount of S-150 on the properties of rubber were tested. For this reason, two groups of experiments were set up. Experimental group 1 (EG 1) was CB reinforced SBR. The ratio of S-150 to rubber was 1:2, and 10–160 phr of CB was divided into 16 gradients with an interval of 10. At the same time, the reference group 1 (RG 1) without S-150 was used as contrast. In RG 1, 10–160 phr of CB was divided into 16 gradients with an interval of 10. Experimental group 2 (EG 2) was WCB reinforced NR. In EG 2, 65 phr of S-150 and 10–160 phr of WCB was divided into 16 gradients with an interval of 10. Meanwhile, the reference group 2 (RG 2) without S-150 was used as contrast. In RG 2, 10–160 phr of WCB was divided into 16 gradients with an interval of 10. The physical performance, tensile properties, and curing characteristics of rubber samples were tested. Moreover, the microstructure of the composite was analyzed by scanning electron microscopy (SEM). The mechanism of increasing reinforcing filler by the addition of S-150 was analyzed according to the like-dissolves-like principle.

## 2. Materials and Methods

### 2.1. Materials

NR (SVR3L) and SBR (1502) were purchased from Dongguan Kunhe plastic Chemical Co. (Dongguan, China), Ltd. CB (N330) and WCB (TS720) was purchased from Shanghai Cabot Co., Ltd. (Shanghai, China). S-150 was purchased from Shanghai Tikem Industrial Co., Ltd. (Shanghai, China). Antioxidant (4010NA), Antideteriorant (M), Zinc oxide (ZnO), Rubber activators (SA), and sulfur (S) were purchased from Nanjing Xinyue Chemical Co., Ltd. (Nanjing, China) Table 1 showed the rubber formulation used in this test. In this rubber formulation, S-150 was used for cross-linking of the rubber compounds. The sulfur curing system consisted of SA, ZnO, and S.

### 2.2. Preparation of Rubber Composites

For the following presentation in this study, initial NR and SBR were represented by rubber, and CB and WCB were represented by filler. In addition to CB and WCB, the other kinds of starting reinforcing filler were defined as reagent. The rubber was cut into pieces by rubber cutter and rolled 3 times by twin-roll mill. The mixer was preheated to 35 °C, the rotational speed was adjusted to 25 r/min, and the rubber after roller pass was put into the mixer for plasticizing for 8 min. The plasticized rubber was rolled 3 min in the twin-roll mill and then rolled, standing for 8 h. Reagent, filler and S were put into 3 beakers. The mixer was preheated to 45 °C, and the speed was adjusted to 35 r/min. The plasticized rubber was placed in the mixer for 2 min and then poured into the reagent. Half of the filler was added and mixed for 4 min. For RG, the second addition of another half of the filler mixing 10 min; for EG, the second addition of the other half filler and S-150 mixing 10 min. Finally, S was added to mix for 5 min to get the mixed gum. After mixing the rubber into the twin-roll mill, it was rolled 5 times, placed in the air for 24 h, and then cured. The mixed rubber was vulcanized on a flat vulcanizer at a curing temperature of 160 °C, curing pressure of 10 MPa, and curing time of 20 min.

During the experimental programs, it could be found that when 100 phr of SBR was bonded with 90 phr of CB in RG 1, the final product would become black granular powder. Therefore, the maximum fraction in RG 1 was 80 phr. Similarly, in RG 2, when 100 phr of NR and 80 phr of WCB were bonded, the final product also became black granular powder. Therefore, the maximum fraction in RG 2 was 70. Meanwhile, in EG 2, when 100 phr of NR were bonded with 140 phr of WCB, the same situation occurred for the final product. Therefore, the maximum fraction in EG 2 was 130 phr, and the cross-sectional vulcanized rubber samples for further detection were shown in Figure 1. The sample with a diameter of 29 mm and thickness of 12.5 mm in Figure 1a was for detection of density, Shore A hardness and rebound resilience according to the requirement of GB/T 1681-2009, and the sample in Figure 1b was for detection of elongation at break, modulus at 100% and tensile strength according to the requirement of GB/T 528-2009.

### 2.3. Characterization

In order to better establish the curing time of rubber, the curing characteristics of rubber was tested by rheometer (GX-LH-2000C, Dongguan Gaoxin Testing Equipment Co., Ltd, Dongguan, China) according to GB/T 16584-1996 ‘Rubber—Measurement of curing characteristics with rotorless curemeters’. Moreover, the density was tested by electronic densitometer of MH-300A according to GB/T 533-2008 ‘Rubber, vulcanized or thermoplastic—Determination of density’; The Shore A hardness was tested by Shaw hardness tester of LX-A according to GB/T 531.1-2008 ‘Rubber, vulcanized or thermoplastic—Determination of indentation hardness—Part 1: Duromerer method (Shore hardness)’; The rebound property of rubber was tested by elastic impact testing machine of WTB-0.5 according to GB/T 1681-2009 ‘Rubber—Determination of rebound resilience of vulcanizates’; The tensile properties were tested by tensile testing machine of WDL-5KN according to GB/T 528-2009 ‘Rubber, vulcanized or thermoplastic—Determination of tensile stress-strain properties’; These 4 equipment were provided by Yangzhou Zhengyi Testing Machinery Co., Ltd, Yangzhou, China.

## 3. Results and Discussion

### 3.1. Curing Characteristics

Normally, the curing temperature of NR and SBR were between 140–180 °C. The optimal curing temperature of NR was 160 °C [43,44,45]. NR compounds were cured at 160 °C and 10 MPa pressure using the optimal curing time determined during rheometric test. Four typical curing systems of RG 1 and EG 1 were selected in this experiment. The curing characteristic parameters of compounds were shown in Table 2. It could be seen from Table 2 that the scorch time of EG 1 (CB-40) compounds was the longest, followed by RG 1 (CB-10) compounds, and the order of positive curing time was EG 1 (CB-40), RG 1 (CB-10), RG 1 (CB-40) and EG 1 (CB-10). △t could characterize the curing rate of the compounds [46]. The curing rate of RG 1 (CB-10) compounds was the lowest, EG 1 (CB-10) compounds was the highest, RG 1 (CB-40) and RG 1 (CB-40) compounds were in the middle and equal. A small amount of S-150 might improve the curing efficiency of the compounds, while excessive addition of S-150 might reduce the curing efficiency of the compounds. The possible reason was that, when a small amount of S-150 was added, the rubber and filler were uniformly dispersed, so the curing rate of the compounds was increased. After adding excessive S-150, a solid-liquid polymer would be formed around the compounds, which increased the viscosity of the compounds. This would cause the cross-linking reaction to be delayed. After adding S-150, M_H_ and ΔM decreased, and t_c90_ time prolonged and the cross-linking degree of the compounds decreased.

The curing characteristics curve of the compounds were shown in Figure 2. After adding S-150, the flat period of the curing characteristic curve of the compounds showed little change. According to the research of other scholars [47,48,49,50], in the study of the curing characteristics of SBR, the curing time was set as 20 min in the positive curing stage. Therefore, the curing temperature of the experiment was set to 20 min.

### 3.2. Analysis of Density and Shore A Hardness

The density of NR was 0.93 g/mL, tge density of SBR was 0.94 g/mL, the density of CB was about 0.375 g/mL, the density of WCB was about 0.2 g/mL, and the S-150 density was about 0.89 g/mL. Effect of filler on the density of rubber samples was investigated and shown in Figure 3. Density was improved gradually by increasing the filler amount. When the filler amount was the same, the rubber density decreased after adding S-150, and the filler amount in the test samples were relatively reduced. The density of EG 1 was lower than that of RG 1 when the CB amount was between 0-80 phr. This might be because the density of S-150 was lower than that of rubber, the ratio of S-150 to rubber in EG 1 was 1:2, and the rubber was relatively reduced. The density of samples in EG 2 were lower than that in RG 2 with corresponding WCB amount in 10-70 phr. The density difference between EG 2 and RG 2 were close to each other with the increase of the amount of WCB. The reason for this was that the constant amount of S-150 in EG 2 was 65 phr, the amount of S-150 in raw rubber decreased relatively with the increase of CB amount. The densities of RG 1 samples were higher than those of RG 2 samples. The main factor was that the density of CB was higher than the density of WCB.

Effect of filler amount on rubber Shore A hardness, as shown in Figure 4. It could be found from Figure 4a that the Shore A hardness of rubber enhanced with the increase of CB amount, and the maximum Shore A hardness was 74 HA when CB amount was 80 phr in RG 1. The maximum Shore A hardness was 95 HA when CB amount was 160 phr in EG 1. The maximum Shore A hardness of EG 1 was 21 HA higher than that of RG 1. After adding S-150, the CB amount in the rubber was relatively reduced and the density of the rubber decreased. Therefore, the Shore A hardness of EG 1 was lower than RG 1. The Shore A hardness difference between EG 1 and RG 1 was closed when CB amount was 0–80 phr. This might be because their CB amount difference was closed.

It could be found from Figure 4b that the Shore A hardness of rubber was enhanced with the increase of WCB amount. The maximum Shore A hardness was 64 HA when the WCB amount was 60 phr in RG 2 and the maximum Shore A hardness was 82 HA when the WCB amount was 130 phr in EG 2. The maximum Shore A hardness of EG 2 was 18 HA higher than that of RG 2. When WCB was 0–70 phr, the Shore A hardness of rubber in EG 2 was higher than that of rubber in RG 2. It might be that, after adding S-150, the WCB amount in the rubber decreased and the density decreased. From Figure 4a,b, it could be found that the type of reinforcing filler and the amount of reinforcing filler were obvious factors for reinforcing Shore A hardness of rubber. S-150 could not be the determinant of rubber Shore A hardness.

### 3.3. Rebound Resilience

Effect of CB on the rebound resilience of rubber samples, as shown in Figure 5. The rebound resilience of rubber began to decrease with the increase of filler. After adding S-150 under the condition of the same filler amount, the rebound resilience of EG 1 was better than that of RG 1 and the rebound resilience of EG 2 was better than that of RG 2. The difference in rebound resilience between EG 1 and RG 1 increased, but the difference between EG 2 and RG 2 decreased with the increase of filler content. This might be because with the increase of filler amount, the CB amount difference between EG 1 and RG 1 increased, and the WCB amount difference between EG 2 and RG 2 decreased. In addition, it was also found that the rebound resilience of RG 1 was lower than that of RG 2, which might be due to the fact that the rebound resilience of NR was better than that of SBR, and that of WCB as reinforcing filler was better than that of CB. There were two factors to improve the rebound resilience of rubber with the addition of S-150. On the one hand, the proportion of rubber increased, and on the other hand, S-150 might change the internal structure of rubber, improve the flexibility of the rubber molecular chain, and reduce the amount of inter-molecular crystallization.

### 3.4. Tensile Properties

Effect of filler on the elongation at break of rubber samples, as shown in Figure 6. Figure 6a,b show that the effects of S-150 on elongation at the break of rubber were similar to that of rebound resilience. The elongation at break in rubber decreased with the increase of filler amount and the elongation at break in rubber increased after adding S-150. This might be because, after the filler amount was added, the agglomeration formed by the adsorbed rubber molecular chain increased, so that the molecular chain that could participate in deformation decreased, the deformation of the molecular chain was limited, and the elongation at break was reduced. After adding S-150, the aggregates formed as a new source of plastic deformation, triggering their own new plastic deformation. The interaction between filler amount and rubber was not perfect, and the interface slip occurred at deformation, thus improving the elongation at break of rubber. It showed that S-150 improved the flexibility of the rubber molecular chain, reduced the number of molecular chain breaks in the process of rubber and filler amount, and improved the elongation at break in rubber. After adding S-150, the elongation at break of rubber increased obviously. It showed that S-150 has positive effect on the elongation at break of rubber.

Effect of filler amount on rubber modulus at 100%, as shown in Figure 7. From Figure 7a,b, it could be found that the modulus at 100% of the rubber decreased after adding S-150. This might be because, after adding S-150, the internal spatial structure of rubber changed, the molecular weight of rubber decreased relatively, the inter-molecular force decreased, and the modulus at 100% of rubber decreased. At the same time, it could be found that the peak value of modulus at 100% moved backward after adding S-150. This might be after adding S-150, the proportion of filler amount was relatively reduced, so the peak moved backward. It could also be found from Figure 7 that the modulus at 100% of RG 1 was larger than RG 2, Meanwhile, EG 1 was smaller than EG 2. This might be because the modulus at 100% of NR was better than SBR. After adding S-150, the EG 1 of the samples was less than modulus at 100% of RG 1. However, when the WCB amount was between 90–130 phr, modulus at 100% of EG 2 was greater than RG 2. This was because the amount of S-150 in EG 1 was higher than that in EG 2 when the filler was more than 70 phr. According to the results of this experiment, it could be found that S-150 has negative effect on modulus at 100% of rubber.

Effect of filler on the tensile strength of rubber samples, as shown in Figure 8. The tensile strength was related to the structure of rubber. When the fraction was small, the secondary valence of inter-molecular interaction was small. Therefore, when the external force was greater than the inter-molecular interaction, inter-molecular sliding would occur and the material would be destroyed. On the contrary, when the molecular weight was large and the inter-molecular force increased, the cohesion of the rubber increased, and the chain segments did not easily slide during stretching, so the damage degree of the material was small.

It was found in Figure 8 that the influence of S-150 on the tensile strength of rubber was similar to modulus at 100%. The tensile strength was related to the molecular structure of rubber. When the molecular weight was smaller, the secondary bond of the interaction between molecules was smaller. Therefore, when the external force was greater than the intermolecular interaction, the material would be destroyed due to the sliding between the molecules. On the contrary, when the molecular weight was higher, the inter-molecular force increased, the cohesion of the compounds increased, and the chain segment did not easily slide during tension, then the damage degree of the material was small. When the filler amount bonded to increase after the peak, the tensile strength would decrease. The tensile strength of the rubber decreased after adding S-150. This might be because the relative molecular weight of the rubber decreased after adding S-150, and the secondary valence of the inter-molecular interaction was small. The chain segments were prone to slide into tensile. Therefore, the tensile strength decreased. It could also be found from the figure that the tensile strength of RG 1 was lower than that of RG 2, and that of EG 1 was lower than that of EG 2. This might be because the tensile strength of NR itself was better than that of SBR. The tensile strength of EG 2 decreased significantly from 50 phr, which might be due to the high proportion of S-150, the relatively low WCB, and the low molecular weight of rubber.

### 3.5. Microscopic

#### 3.5.1. Microscopic Analysis of CB Reinforced SBR

In order to better determine the effect of S-150 on rubber, rubbers reinforced with different amounts of CB were observed by SEM. The test results were shown in Figure 9. It could be found from Figure 9 that the particle size of RG 1 was smaller than that of EG 1, the surface was smooth, the aggregates were uniform, and no bubbles were generated. This might be due to the improvement of the plasticity of rubber with the addition of S-150, and the melting of rubber and S-150 to form solid-liquid rubber, which increased the contact surface area between rubber and filler. This might be due to the improvement of rubber plasticity after adding S-150. Rubber and S-150 were melted to form solid-liquid rubber, which increased the contact surface area between rubber and filler. When the same CB was added, the amount of CB in the samples of EG 1 was relatively less, the curing degree of rubber was higher, and the aggregate particles were less. The size of aggregates varies with different CB in RG 1 with the increase of CB, which might be caused by the uneven binding of CB amount and raw rubber in the mixing process. The aggregates increased obviously with the increase of CB In EG 2, the aggregates were more uniform, and there were no large aggregates, which indicated that EG 2 still had some binding space between rubber and filler.

When the CB amount reached 90 phr, the aggregates with more CB particles began to break up and form powder or fragments. The crumbling powder in rubber became more and more pronounced with the extension of mixing time, the rubber amount in the mixer became less and less, until finally the rubber collapsed. When the amount of CB reached 90 phr, the disintegration state of carbon black was reached, as shown in Figure 10.

It could be found that the size of CB particles after collapse was different, because the aggregate size of CB formed after the amount of rubber and CB particles reached a certain degree. The larger molecular weight between aggregates would produce a larger repulsion force, breaking the rubber molecular chain. Another possibility was that during the mixing process, the shear force provided by the mixer cut the large mixing rubber into small mixing rubber. When small pieces of rubber came into contact with CB, the surface area was relatively large. When the combination of small pieces of rubber and filler exceeded the upper limit, the small pieces of mixed rubber began to collapse. The powder and carbon black formed by the collapse of small pieces of rubber combined with the surrounding pieces of rubber, finally forming rubber powder particles of different sizes, as shown in Figure 10.

The section specimen samples of SBR reinforced with 90–160 phr of CB is shown in Figure 11. It can be seen from Figure 11 that, after the CB amount was more than 90 phr, the number of CB aggregates in the rubber was large. Meanwhile, the dispersion was uniform. The size of the aggregates was close, the large particle aggregates were less, and the spatial structure was relatively stable. When the amount of CB was 130 phr and 150 phr, the reticulation space of rubber was basically filled with CB aggregates, and the small aggregate particles began to bind with each other to form a gully line. It was similar to CB with 70 phr, so it could be judged that the rubber was basically close to the upper limit.

#### 3.5.2. Morphological Analysis of WCB Reinforced NR

Figure 12 showed that the internal structure of RG 2 was similar to RG 1. Meanwhile, EG 2 was similar to EG 1. EG 2 aggregates were relatively uniform, with fewer large particle aggregates without pores and bubbles. There were more irregular aggregates in RG 2 and larger aggregates in rubber. By comparison, it was found that the internal structure of EG 2 was significantly better than that of RG 2. When WCB bonded with rubber, through uniform dispersion, most of the specific surface area had strong interaction with the contact interface of rubber. This interaction was not only closely related to improving the mechanical properties of vulcanized rubber, but also affected the formation of rubber-filled gel, the expansion behavior, the mobility of polymer chain molecules, the glass transition temperature, and the viscosity [51,52]. Usually, when the WCB was mixed like glue, the viscosity of the glue increased, which would cause many problems during processing, so many methods to reduce viscosity have been studied. After adding S-150, the problem of uneven dispersion of WCB was effectively solved.

The section specimen samples of NR reinforced with 90 and 120 phr of WCB are shown in Figure 13. It can be found from Figure 13 that, when the WCB amount was 90 phr, the size of aggregates in rubber was closed, and the network structure was relatively uniform. When the WCB was 120 phr, the uniform WCB aggregates formed a relatively obvious gully space inside the rubber. According to the analysis results of 3.4.1, it was a sign that the amount of rubber and filler had reached their upper limits. Adding a lot of WCB would cause the effects shown in Figure 10 to occur.

### 3.6. Mechanism Analysis

NR was cut into a rubber block and placed in S-150 for 12 h to observe the results, as shown in Figure 14. It could be found from Figure 14a that the density of rubber was higher than that of S-150. Figure 14b shows that the physical and chemical reaction of rubber in S-150 generated a hairy edge and the volume extended horizontally. It can be seen in Figure 14c,d that there were different changes in the 1 and 2 sides of the rubber. This might be the 2 side was subjected to its own pressure, and only the rubber at the edge of the 2 side reacted with S-150. Therefore, the lateral extension of the rubber block was larger than the longitudinal extension.

It can be seen in Figure 14 that the rubber soaked in S-150 for 24 h consisted of three parts: liquid rubber (p), solid-liquid blending rubber (q), and original rubber (r). After the rubber block was rolled 3 times in the open mill, a smaller fragment was formed. Then the mixer provided shear force in the mixing process, so that smaller fragments could make full contact with S-150, thus providing a favorable condition for the rubber to change from p state to q state and r state. When the mixing temperature increased from 45 °C to 50 °C, the thermal expansion of the rubber and the activity of S-150 increased, so the reaction rate increased. The internal network structure space of the rubber increased, the flexibility of the rubber macro-molecular chain increased, the contact area between the rubber and filler particles increased, and the number of filler particles that could be accommodated increased, so the amount of rubber and filler binding increased.

According to the like-dissolves-like principle, when the properties between solute and solvent are similar, solute is more likely to be dissolved in the extraction solvent to produce obvious extraction phenomena [53,54]. S-150 was a kind of solvent oil. Its properties were similar to those of gasoline. Colorless liquid belongs to the non-polar solvent class. Its main components were preponed and isopropylbenzene (_C9H12_), 1,3,5-trimethylbenzene (_C9H12_), xylene and its isomers (_C8H10_), 1,2,4-trimethylbenzene (_C9H12_) dissolved in ethanol and ether. Organic small molecules with non-polar functional groups such as me-thyl (−CH_3_) in NR and methylene (−CH2−) in SBR. Non-polar solvents with weak polarity were more easily compatible with organic small molecules.

Figure 15 shows the schematic diagram of S-150 reinforcing rubber principle. Rubber bonded with filler was RG. Rubber and filler bonded with S-150 was EG. After adding S-150, the internal network space of the rubber was expanded, the flexibility of the rubber molecular chain was improved, and the macro-molecular chain nodes were reduced. Therefore, physical and chemical reactions with more filler could occur. On the other hand, the internal network space of rubber was expanded, and the contact surface area between filler particles and rubber molecules were increased, the internal network space of rubber was expanded, and the contact surface area between filler particles and rubber molecules were increased, which allowed the cross-linking reaction between rubber and filler to improve curing efficiency.

## 4. Conclusions

This study established the feasibility of adding S-150 to a rubber formula. According to the like-dissolves-like principle, rubber and S-150 could be integrated with each other. The internal network space of rubber increased, the molecular chain was stretched, and the nodes between the molecular chains were reduced. This meant that rubber could be bonded with more filler to achieve the purpose of strengthening rubber properties. By analyzing SEM images showing the rubber bonding with small amounts of filler, it was found that the filler inside the rubber was evenly dispersed, the size of the aggregates was similar, and there were no bubbles or pores. When rubber was bonded with a large amount of filler, the particle size of the rubber internal aggregates was also uniform, and the net structure space showed certain regularity.

The results showed that the curing efficiency of the compounds was improved after adding S-150. The density of S-150 was small, and the overall density of rubber decreased with the addition of S-150. Although S-150 had little effect on the Shore A hardness of rubber, the binding amount of rubber and filler was increased after adding S-150, so as to achieve the purpose of improving Shore A hardness in rubber. S-150 has positive effect on the rebound resilience and elongation at break of rubber. The rebound resilience and elongation at break also increased with the increase of the amount of S-150. S-150 has a negative effect on modulus at 100% and on the tensile strength of rubber. When the amount of S-150 in rubber was low and the amount of filler was high, the modulus at 100% and the tensile strength of rubber would be improved. It can be predicted that S-150 could be applied to the study of more kinds of rubber.

## Figures and Tables

**Figure 1 polymers-14-02745-f001:**
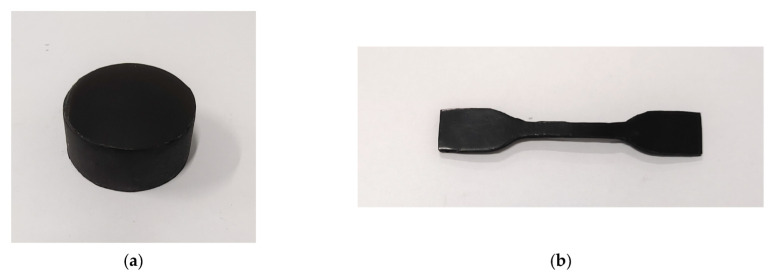
The cross-sectional vulcanized rubber samples for further detection. (**a**) Sample with diameter of 29 mm and thickness of 12.5 mm for detection of density, Shore A hardness and rebound resilience; (**b**) Samples for detection of elongation at break, modulus at 100% and tensile strength.

**Figure 2 polymers-14-02745-f002:**
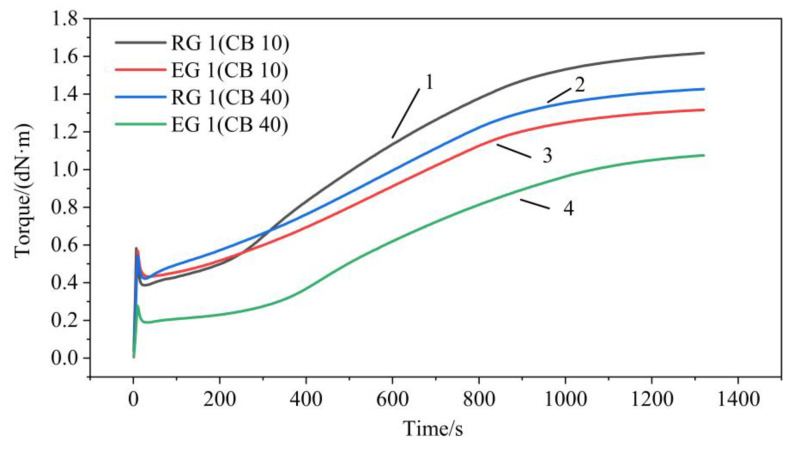
Curing characteristics of rubber compounds with different curing systems. Line 1 was for RG 1 with CB of 10 phr; Line 2 was for EG 1 with CB of 10 phr; Line 3 was for RG 1 with CB of 40 phr; Line 4 was for EG 1with CB of 40 phr.

**Figure 3 polymers-14-02745-f003:**
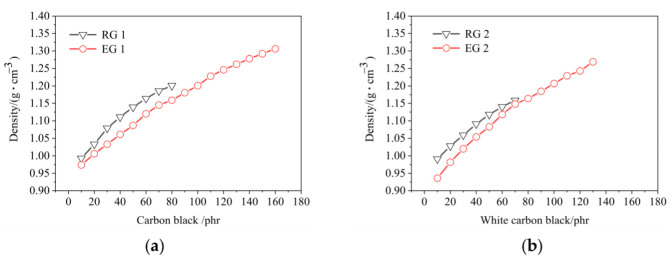
Effect of filler on the density of rubber samples. RG 1 and EG 1 were for SBR rubber mixtures; RG 2 and EG 2 were rubber mixtures based on NR rubber. (**a**) Effect of CB on the density of rubber samples; (**b**) Effect of WCB on the density of rubber samples.

**Figure 4 polymers-14-02745-f004:**
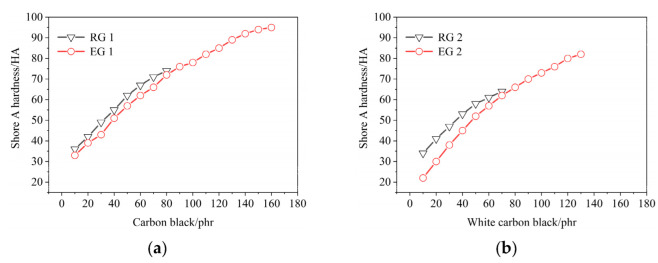
Effect of filler on the Shore A hardness of rubber samples. RG 1 and EG 1 were for SBR rubber mixtures; RG 2 and EG 2 were rubber mixtures based on NR rubber. (**a**) Effect of CB on the Shore A hardness of rubber samples; (**b**) Effect of WCB on the Shore A hardness of rubber samples.

**Figure 5 polymers-14-02745-f005:**
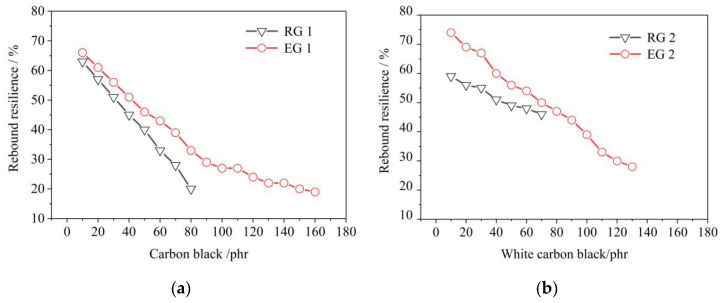
Effect of filler on the rebound resilience of rubber samples. RG 1 and EG 1 were for SBR rubber mixtures; RG 2 and EG 2 were rubber mixtures based on NR rubber. (**a**) Effect of CB on the rebound resilience of rubber samples; (**b**) Effect of WCB on the rebound resilience of rubber samples.

**Figure 6 polymers-14-02745-f006:**
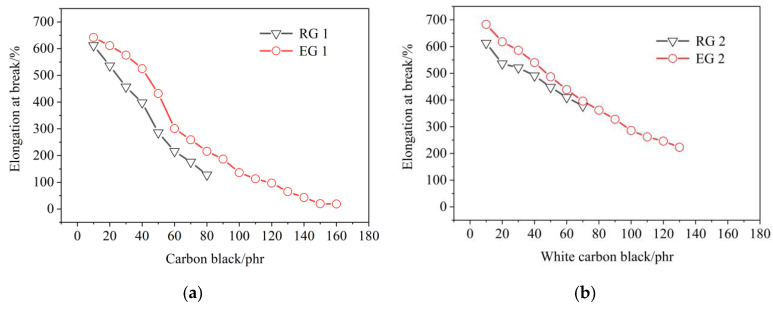
Effect of filler on the elongation at break of rubber samples. RG 1 and EG 1 were for SBR rubber mixtures; RG 2 and EG 2 were rubber mixtures based on NR rubber. (**a**) Effect of CB on the elongation at break of rubber samples; (**b**) Effect of WCB on the elongation at break of rubber samples.

**Figure 7 polymers-14-02745-f007:**
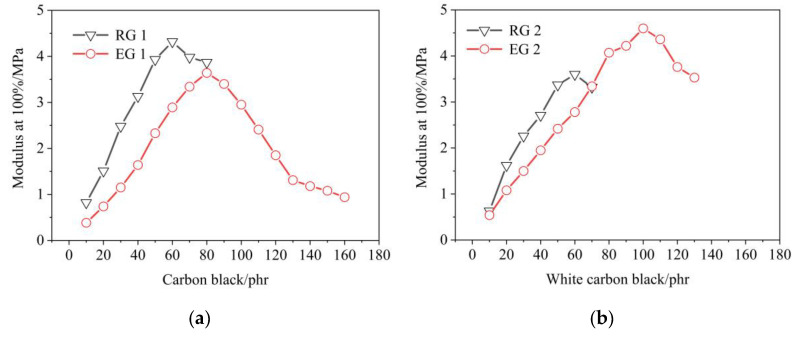
Effect of filler on the modulus at 100% of rubber samples. RG 1 and EG 1 were for SBR rubber mixtures; RG 2 and EG 2 were rubber mixtures based on NR rubber. (**a**) Effect of CB on the modulus at 100% of rubber samples; (**b**) Effect of WCB on the modulus at 100% of rubber samples.

**Figure 8 polymers-14-02745-f008:**
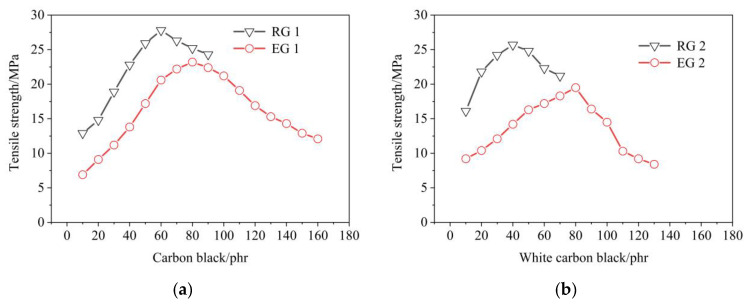
Effect of filler on the tensile strength of rubber samples. RG 1 and EG 1 were for SBR rubber mixtures; RG 2 and EG 2 were rubber mixtures based on NR rubber. (**a**) Effect of CB on the tensile strength of rubber samples; (**b**) Effect of WCB on the tensile strength of rubber samples.

**Figure 9 polymers-14-02745-f009:**
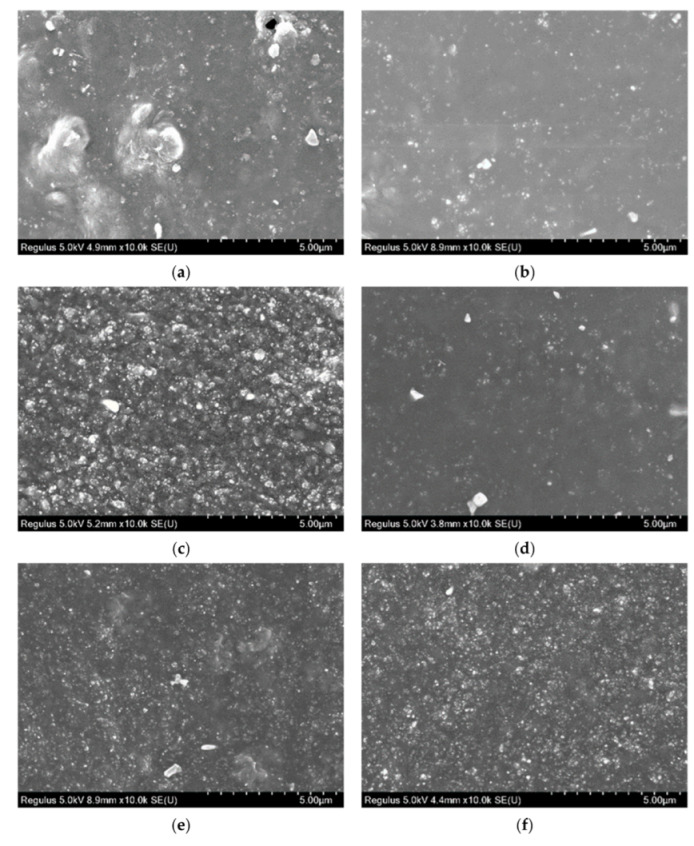
SEM images of SBR section specimen (CB 0-70 phr, magnification 10,000×). (**a**) RG 1–10 phr; (**b**) EG 1–10 phr; (**c**) RG 1–30 phr; (**d**) EG 1–30 phr; (**e**) RG 1–50 phr; (**f**) EG 1–50 phr; (**g**) RG 1–70 phr; (**h**) EG 2–70 phr.

**Figure 10 polymers-14-02745-f010:**
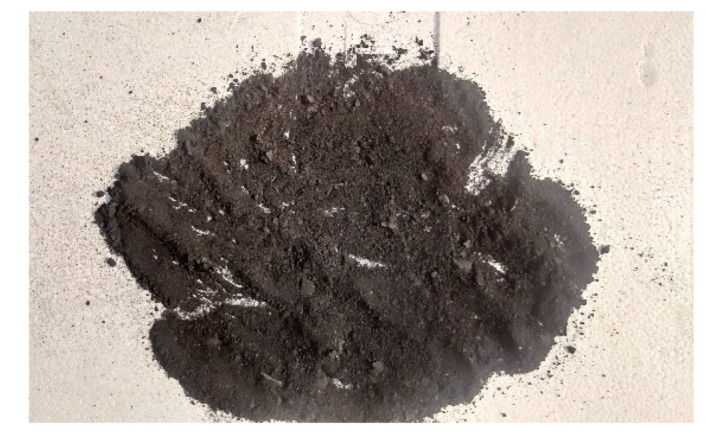
90 phr of CB powder formed in the mixing process.

**Figure 11 polymers-14-02745-f011:**
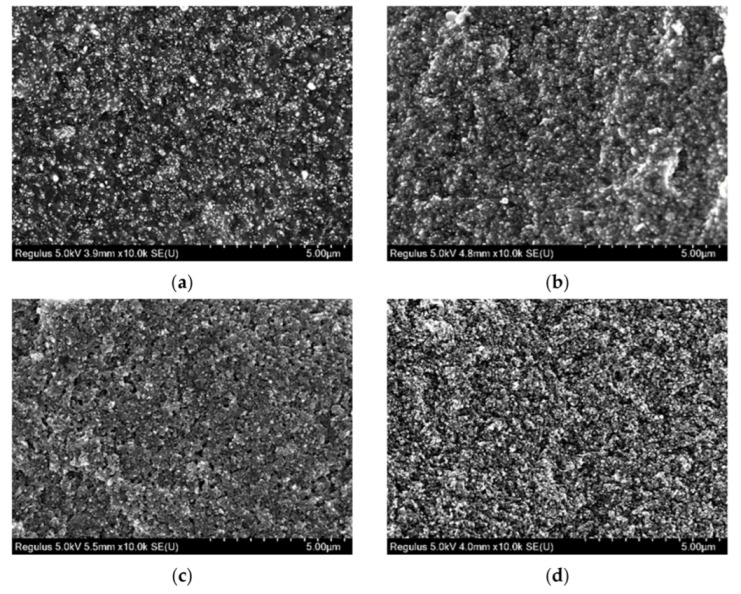
SEM images of SBR section specimen (CB 90–150 phr, magnification 10,000×). (**a**) EG 1–90 phr; (**b**) EG 1–110 phr; (**c**) EG 1–130 phr; (**d**) EG 1–150 phr.

**Figure 12 polymers-14-02745-f012:**
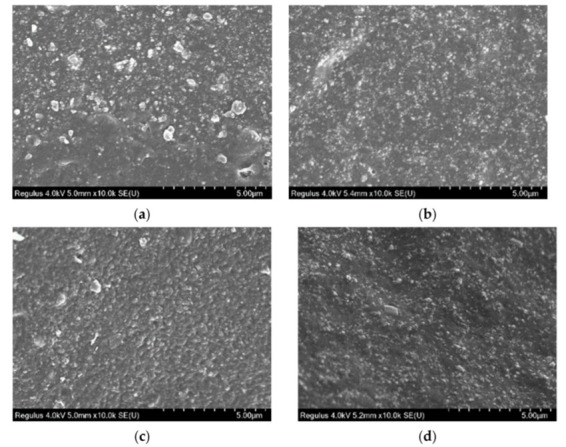
SEM images of NR section specimen (WCB 0–60 phr, magnification 10,000×). (**a**) RG 2–30 phr; (**b**) EG 2–30 phr; (**c**) RG 2–60 phr; (**d**) EG 2–60 phr.

**Figure 13 polymers-14-02745-f013:**
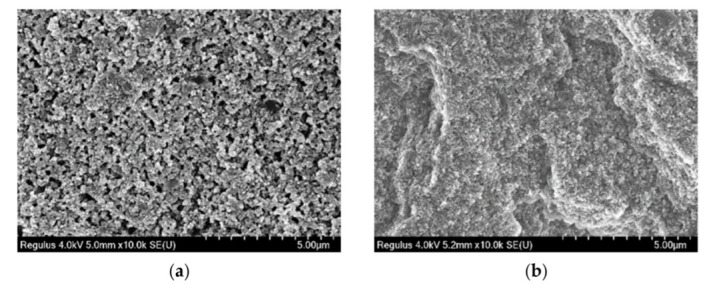
SEM images of NR section specimen (WCB 90–120 phr, magnification 10,000×). (**a**) EG 2–90 phr; (**b**) EG 2–120 phr.

**Figure 14 polymers-14-02745-f014:**
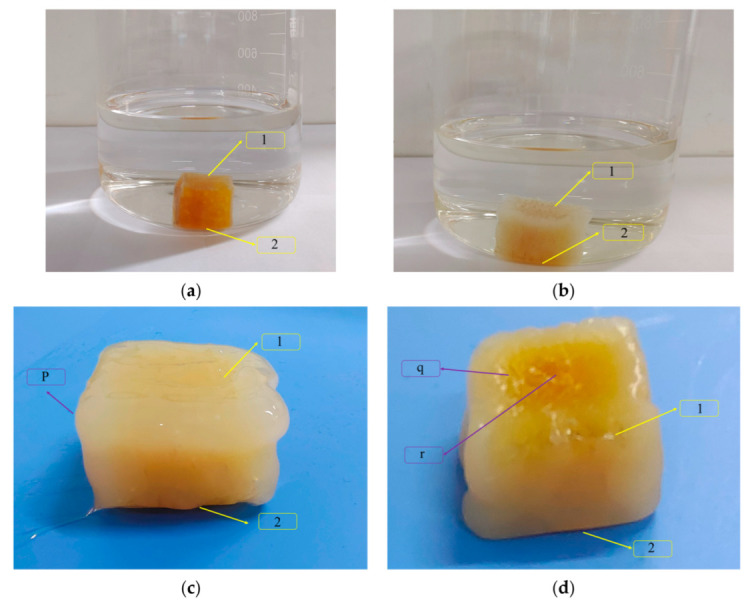
Soak NR in S-150. (**a**) Cube rubber block bubble in the case of S-150, volume 1.5 × 1.5 × 1.5 cm^3^; (**b**) The rubber block has a volume of 1.8 × 2 × 2 cm^3^ in S-150 for 4 h; (**c**) When the rubber block was removed from S-150 for 12 h, the volume was 2 × 2.5 × 2.5 cm^3^; (**d**) Rubber block reversed. Here 1 was the top area of the rubber and 2 was the bottom area of the rubber; p, q and r were the liquid rubber, solid-liquid blending rubber and original rubber respectively.

**Figure 15 polymers-14-02745-f015:**
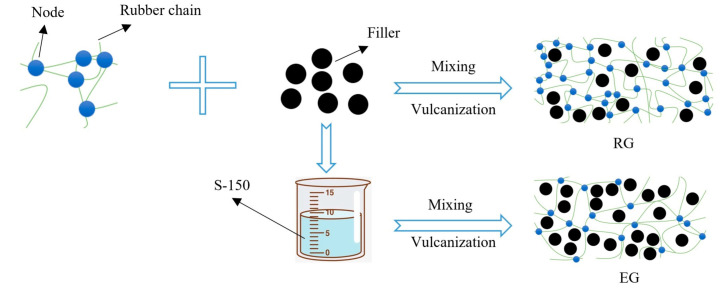
The schematic diagram of S-150 reinforcing rubber principle.

**Table 1 polymers-14-02745-t001:** Rubber formulation.

Materials	RG 1	EG 1	RG 2	EG 2
SBR	100	100	-	-
NR	-	-	100	100
CB	10–160	10–160	-	-
WCB	-	-	10–160	10–160
S-150	-	5–80	-	65
4010NA	2	2	2	2
M	1.8	1.8	1.8	1.8
ZnO	5	5	5	5
SA	3	3	3	3
S	3	3	3	3

**Table 2 polymers-14-02745-t002:** Curing characteristic parameters.

Curing Characteristics	M_L_/(dN·m)	M_H_/(dN·m)	△M	tc10/min	tc90/min	Δt
RG 1 (CB-10)	0.38	1.61	1.23	3.4	15.3	11.9
EG 1 (CB-10)	0.43	1.42	0.99	2.1	16.6	14.5
RG 1 (CB-40)	0.42	1.32	0.9	2.4	15.9	13.5
EG 1 (CB-40)	0.21	1.08	0.87	5.5	17.6	12.1

M_L_—minimum torque; M_H_—maximum torque; t_c10_—Initial curing time; t_c90_—positive curing time. ΔM—MH- ML. Δt—t_c90_-t_c10_.

## Data Availability

The data that support the findings of this study are available from the corresponding author upon reasonable request.

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
