# Peer review of "Study on Increasing the Binding Amount of Rubber and Reinforcing Filler by Adding Aromatic Solvent Oil"

_polymers, 2022, doi:10.3390/polym14132745_

Round 1
Reviewer 1 Report
The manuscript “Study on Increasing the Binding Amount of Rubber and Rein-2 forcing Filler by Adding Aromatic Solvent Oil” refers to properties of rubber. The topic of the publication it is not although it has practical importance especially for the rubber companies working in the area of elastomeric materials. I find this manuscript interesting and worth publishing. However there are comments for Authors which I would like to point before accepting that work. So my recommendation is accept after minor revision.

Author Response
Response to reviewer 1
Thank you very much for your kind review to our manuscript and positive assessment to our research. We have revised the manuscript carefully according to your and other reviewers’ comments. The responses to your comments are as follows.
- Abstract. Could authors add to abstract some numerical values confirming the conclusions, such as the values of increased rebound resilience. Some spelling mistakes should be improved: line 14 (Characterized -characterized)
Response:
Thank you very much for your kind suggestion. We have added some parameters about the rebound resilience and elongation at break. Some spelling mistakes have been improved and these modifications are highlighted in yellow in the revised manuscript.
- The introduction is written concisely, all aspects and problems connected with studied topic are described in details , the amount of references is sufficient. There are some language mistakes which I pointed below. - Line 67 “ Marcin et al [35] should be Masłowski et al. , see Reference 35 -Marcin, Justyna, Krzysztof there are names. Marcin is similar to English names like Martin so in reference should be Masłowski M. Miedzianowska J., Strzelec K.
Response:
Thank you very much for your kind suggestion. Relevant references have been revised and these modifications were highlighted in yellow.
- Materials and methods part is written adequately.
Response: Thank you very much for your kind suggestion.
- Results and discussion-Vulcanization Characteristic. Line 58 Figure 2. The colours on plots are not in agreement with the description below the plot. The description is as follows: Figure 2. Vulcanization characteristics of CB compounds with different Vulcanization systems. 158 Line 1 is for RG 1 with CB of 10 phr; Line 2 is for EG 1 with CB of 10 phr; Line 3 is for RG 1 with CB of 40 phr; Line 4 is for EG 1with CB of 40 phr. And according to the plot colours the Line 1 is RG 1 CB 10 phr (black), Line 2 RG 1 CB 40 phr (blue), Line 3 EG 1 CB 10 phr (red), Line 4 EG 1 CB 10 phr (green). Please, correct the description below the plot.
Could you add to this part to the text the values from the vulcanization plots such as the values of the minimum torque, maximum torque, torque=torque(maximum)-torque(minimum) confirming the observations. - -
Could you underlined in the text that this plots are for SBR rubber, e.g. add in line 154 four typical SBR vulcanization system . Could Authors add some typical plots for the NR mixtures showing influence of S-150 on the kinetics of curing for NR rubber?
- Line 165 “After adding S-150, the torque of the com-165 pound decreased and the positive vulcanization time also decreased. This might be the 166 mixture was more uniform and the vulcanization efficiency was improved.” I cannot agree with statement that the vulcanization efficiency was improved. Did Authors studied the C.R.I index (cure rate index)? The added values of C.R.I index for studied rubbery mixtures could improve the evaluability of the manuscript.
Response:
Thank you very much for your kind suggestion. Figure 2. has been corrected. Curing characteristic parameters have been added in Table 2. The part about vulcanization efficiency in the paper has been revised and and these modifications are highlighted in yellow in the revised manuscript.
- Results and discussion-Analysis of density and Shore A Hardness
Line 180: Should be The density difference between EG 2 and 180 RG 2 was closed to each other with the increase of the amount of WCB. Similar in Line 183.
Line 184 should be The main factor was that the density of CB was higher than the density of WCB.
Could Authors add to the Figure 3 and Figure 4 the description that the plots RG 1 and EG1 are for SBR rubber mixtures and plots RG2 and EG2 are rubber mixtures based on NR rubber.
Response:
Thank you very much for your advice. Line 180:CB has been changed to WCB. Line 184:CB has been changed to WCB. SBR and NR mixtures have been added. These modifications are highlighted in yellow in the revised manuscript.
- Results and discussion - Rebound resilience, Tensile properties.
Could Authors add to the Figure 5 and Figure 6, 7, 8 the description that the plots RG 1 and EG1 are for SBR rubber mixtures and plots RG2 and EG2 are rubber mixtures based on NR rubber.
-Line 242 should be “it showed that S-150”
-The explanation of the obtained results in this part of manuscript is correct but I have one comment, moderate English changes are required to improve this part of the manuscript.
Response:
Thank you very much for your kind suggestion. Figure 5 and Figure 6, 7, 8 have been added. Line 242 has been modified. English has been revised. These modifications are highlighted in yellow in the revised manuscript.
- Results and discussion - Micromorphology.
Line 292 – please unify and use in whole text S-150.
-I agree with the explanation of the obtained results in this part of manuscript but please go once again through this part of manuscript as there are some spelling/language mistakes. Also I would recommend to re-check the grammar (a, the).
Response:
Thank you very much for your kind suggestion. Smur150 has been uniformly used in whole text. The relevant grammar has been modified and these modifications are highlighted in yellow in the revised manuscript.
- Results and discussion – Conclusions.
The explanation of the obtained results in this part of manuscript is correct.
-Moderate English changes are required to improve this part of the manuscript.
Response:
Thank you very much for your kind suggestion. The relevant grammar has been modified and these modifications are highlighted in yellow in the revised manuscript.

Reviewer 2 Report
Dear Authors,
This paper represents the effects of aromatic oil to the filler dispersion in the rubber matrix and the mechanical properties. Authors have well described the experimental results and I support for publication after major revision as follows
1. What is rubber adhesive (SA)? What is the name of the vulcanizing accelerator(s) used to cure rubber? What are the physical and chemical properties of Aromatic solvent oil (S-150)?
2. “The mixed rubber was vulcanized on a flat vulcanizer under the vulcanization temperature of 160 °C, vulcanization pressure of 10 MPa, and vulcanization time of 20 min”. If you used optimum curing time then above line should be modified. If you used same curing time for all the compounds then put the reason.
3. Please discuss vulcanization characteristics with including the cure rate index and the true rate of vulcanization for more logical explanations. This paper could be helpful (Polymers 2022, 14(9), 1921).
4. Density values included should be checked very carefully. I think these are not correct.
5. You used the term cross-linking density to explain mechanical properties. However I did not find the measurement of cross-link density in the text.
6. The effect of other processing oil on filled rubber compounds should be properly cited. e.g. (J. Appl. Polym. Sci., 131, doi: 10.1002/app.39988).
7. Some typos should be checked carefully.
Author Response
Response to reviewer 2
Thank you very much for your kind review to our manuscript and positive assessment to our research. We have revised the manuscript carefully according to your and other reviewers’ comments. The responses to your comments are as follows.
- What is rubber adhesive (SA)? What is the name of the vulcanizing accelerator(s) used to cure rubber? What are the physical and chemical properties of Aromatic solvent oil (S-150)?
Response:
Thank you very much for your kind suggestion. SA is a widely used vulcanization active agent in natural rubber, synthetic rubber and latex, and can also be used as plasticizer and softening agent. The name of the vulcanizing accelerator(s) used to cure rubber is Sulphur. Aromatic solvent oil has excellent solubility and chemical and physical stability, colorless liquid, insoluble in water, high boiling point, slow volatilization and good leveling, so it can be used as a solvent for all kinds of coatings.
- “The mixed rubber was vulcanized on a flat vulcanizer under the vulcanization temperature of 160 °C, vulcanization pressure of 10 MPa, and vulcanization time of 20 min”. If you used optimum curing time then above line should be modified. If you used same curing time for all the compounds then put the reason.
Response:
Thank you very much for your kind suggestion. The positive vulcanization time of rubber was a stage, and the test samples were in the positive vulcanization stage in 20 min. Therefore, it was reasonable to set the curing time of all compounds as 20 min.
- Please discuss vulcanization characteristics with including the cure rate index and the true rate of vulcanization for more logical explanations. This paper could be helpful (Polymers 2022, 14(9), 1921).
Response:
Thank you very much for your kind suggestion. The reference was added in the text according to your suggestion in the section 3.1 and the added part was highlighted in yellow. Meanwhile,we discussed vulcanization characteristics with including the cure rate index and the true rate of vulcanization.The test of rheometric was to determine the positive curing time, and the focus of this paper was the effect of S-150 on rubber.
- Density values included should be checked very carefully. I think these are not correct.
Response:
Thank you very much for your kind suggestion. The density value has been modified and these modifications were highlighted in yellow.
- You used the term cross-linking density to explain mechanical properties. However, I did not find the measurement of cross-link density in the text.
Response:
Thank you very much for your kind suggestion. The mechanical properties were explained by other ways instead of cross-linking density. The related presentations were added in the revised manuscript, and these modifications were highlighted in yellow.
- The effect of other processing oil on filled rubber compounds should be properly cited. e.g. (J. Appl. Polym. Sci., 131, doi: 10.1002/app.39988).
Response:
Thank you very much for your kind suggestion. Reference 10 was replaced with this reference.
- Some typos should be checked carefully.
Response:
Thank you very much for your kind suggestion. The relevant typos have been modified and these modifications were highlighted in yellow.

Round 2
Reviewer 2 Report
The paper is well improved.